# A Low-Power Communication Strategy for Terminal Sensors in Power Status Monitoring

**DOI:** 10.3390/s25051317

**Published:** 2025-02-21

**Authors:** Qingqing Wu, Yufei Wang, Di Zhai, Yang Lu, Cheng Zhong, Yihan Liu, Yuxuan Li

**Affiliations:** 1China Electric Power Research Institute, Beijing 102209, China; wqqing1231@163.com (Q.W.); wallyful@126.com (Y.W.); dasluogu@163.com (D.Z.); luyangnnu@126.com (Y.L.); 2State Grid Xiongan New Area Electric Power Supply Company, Xiongan 071000, China; hebeizhongc@163.com; 3College of Information and Electrical Engineering, China Agricultural University, Beijing 100083, China; 4School of Electrical and Electronic Engineering, North China Electric Power University, Beijing 102209, China; 120222201317@ncepu.edu.cn

**Keywords:** power pipe galleries, status monitoring, radio frequency energy harvesting, low-power communication, cognitive backscatter, collision probability

## Abstract

The widespread application of terminal sensors in power pipe galleries (PPGs) has significantly improved our ability to monitor power equipment status. However, the difficulties in battery replacement caused by confined space and energy loss caused by communication conflicts between sensors due to existing low-power communication strategies results in a lack of reliable energy supply for terminal sensors. In this context, a low-power communication strategy for terminal sensors is proposed. Firstly, a demand analysis is conducted on the status monitoring of PPGs, and a technical framework for low-power communication of terminal sensors is proposed. Afterward, a system model for the low-power communication of terminal sensors is established based on cognitive backscatter technology. Subsequently, key technologies, such as RF energy harvesting of terminal sensors and distance–energy level coupling analysis, are proposed to achieve continuous energy supply and avoid communication conflicts in the system model. Finally, a wireless communication simulation environment for PPGs is established to simulate the status monitoring process, based on terminal sensors, in order to verify the effectiveness of the proposed method.

## 1. Introduction

The operation, maintenance, and monitoring of power equipment is a crucial link in realizing the characteristics of a new power system, such as flexibility, intelligence, supply–demand coordination, and safety sufficiency [1,2,3,4]. Power pipe galleries (PPGs) are generally located underground and are used for transmitting electric energy. Existing research results indicate that zero-power Internet of Things (IoT) technology can achieve good complementarity with existing IoT communication technologies and will be a necessary path for the development of the next generation of IoT technology [5,6,7,8]. Currently, there are numerous terminal sensors in scenarios such as PPGs, transmission lines, and substations [9,10], and their working environments is complex. Terminal sensors are usually installed near primary equipment, making it difficult to acquire reliable power nearby. Consequently, the sensors rely on battery power supply, which needs to be replaced by staff in hazardous environments with strong electromagnetic interference and high temperature and humidity, making daily maintenance difficult [11]. Therefore, studying low-power communication strategies for terminal sensors contributes to the monitoring capability of power equipment in complex operating conditions.

At present, wireless sensor networks can meet the needs of online status-monitoring services for various power grids and equipment and provide a simple deployment solution for sensors without cables. Many studies have been devoted to developing wireless sensor power consumption [12,13]. The current research on low-power sensors mainly focuses on RF energy harvesting and backscatter technology. In terms of RF energy harvesting, ref. [14] explores the application of sensor-aided zero-energy reconfigurable intelligent surfaces (SAZE-RISs) within the integrated sensing, communication, and powering (ISCAP) framework. SAZE-RIS provides an energy-efficient solution for ISCAP by meeting the needs of end-users and by powering reconfigurable intelligent surfaces. Reference [15] proposes a combined design method for radio frequency (RF) energy harvesting and management to obtain the required output parameters of the radio frequency energy harvester, thus enabling load operation at the design stage. The proposed WSN can achieve self-powered operation at a distance of 13.5 m from a 27 dBm RF energy source. Reference [16] proposed a microscale RF energy harvesting and power management solution to reduce the start-up power of energy harvesting and expand the energy harvesting range. References [17,18] provide a detailed introduction to the working principles and optimization methods of antennas, matching networks, and rectification circuits in RF energy harvesting systems, analyze their application levels, and discuss the current problems and development trends. In terms of backscatter systems, Reference [19], by optimizing power allocation, time allocation, the reflection coefficient, and the energy distribution coefficient, considering the constraints of the maximum RF source transmission power and the minimum circuit power consumption, the sum of the backscattered data rate and the information transmission rate can be maximized. References [20,21,22] introduce the evolution of existing backscatter communication architectures, including single-site, dual-site, environmental, parasitic, mutualistic symbiotic, and mutualistic symbiotic architectures that integrate active and passive transmission. Reference [23] proposes that cognitive backscatter technology strongly supports low-power IoT technology. The system throughput is maximized by jointly optimizing secondary transmitters’ parameters, considering the IoT node interruption rate and minimum energy collection constraints. In [24,25], multiple backscatter devices communicate using a protocol that integrates Non-Orthogonal Multiple Access Technology (NOMA) and Dynamic Time Division Multiple Access Technology to improve system throughput and reduce system power consumption.

However, the above existing achievements do not necessarily support the passive operation of terminal sensors for equipment status monitoring. On the one hand, the way to distinguish users based on technologies such as support vector machine, NOMA, time-division multiple access, and carrier sense multiple access (CSMA) usually requires built-in batteries or RF sources to provide large and sustained energy for backscatter devices. On the other hand, existing research based on RF energy harvesting and backscatter technology mostly focuses on performance optimization dimensions such as expanding the energy harvesting range, improving system throughput, and reducing system power consumption, but there is no further discussion on the combination of these technologies with power application scenarios.

Based on the above issues, this article proposes a low-power communication strategy based on cognitive backscatter communication, which draws on the experience of cognitive radio technology in low-power communication systems for PPGs scenarios. This strategy utilizes RF energy harvesting technology to provide reliable energy supply for passive terminal sensors and transmits state-sensing information based on cognitive backscatter technology. The effectiveness is verified through simulation experiments. The main innovative points are as follows:Combined with the demand analysis of PPGs, a technical framework for low-power communication of terminal sensors is proposed to explain the overall research ideas of this paper.A cognitive backscatter low-power communication system model was proposed, which includes three stages: the passive sensing node RF energy harvesting and information transmission stage, the RF source information transmission stage, and the passive sensing node information transmission stage. This model provides a solution to avoid the problem of co-frequency interference between the power supply and the communication of the low-power terminal sensor.Given that the research object of this article is terminal sensors with low-power consumption and small sensing data volume, key technologies such as RF energy harvesting, distance–energy level coupling, and wake-up delay correlation analysis methods are proposed to reduce the transmission collision probability of low-power communication systems with cognitive backscatter.

## 2. Research Ideas

### 2.1. Requirement Analysis

Currently, with the rapid developments in the power IoT, the number of narrowband IoT sensing terminal accesses has significantly increased. The terminal sensing nodes present characteristics such as a wide variety, large quantity, complex operating environment, wide distribution range, and significant environmental differences. The problems of high power consumption, difficulty in obtaining electricity, and complex installation of communication cables in existing IoT sensing technologies are becoming increasingly prominent, making it difficult to meet the full field-scene sensing needs of power systems [26,27]. Among them, the scene of PPGs has a certain representativeness, and it faces a series of technical bottlenecks as follows:Online monitoring devices for PPGs need a power supply for operation, but most channels lack a power supply and cannot be deployed on a large scale. The embedded lithium battery has a lifespan of about 1–2 years and needs to be replaced regularly.At present, cable equipment and cable channel monitoring mainly rely on wired methods, such as cable-fiber-distributed temperature measurement, infrared monitoring, etc. Communication cable laying is complex.

In response to the above technical bottlenecks, there is an urgent need for IoT sensing capabilities with lower power consumption and lower operational costs to enhance the reliability of power grid operation and support the construction of new power systems.

### 2.2. Technical Framework

Based on the previous requirements analysis, this article proposes a low-power communication strategy based on cognitive backscatter. Corresponding to the two technical bottlenecks faced by the current PPGs summarized in Section 2.1, the following settings are made for the strategy:

Setting 1. Use active sensors as RF sources to provide RF energy to various terminal passive sensors for their own perception, signal processing, and communication.

Setting 2. Given the characteristics of limited energy and the small amount of perceived data of passive sensors, distance-level coupling analysis and other methods are adopted in low-power communication systems based on cognitive backscatter to avoid communication conflicts.

Specifically, taking the scenario of PPGs status monitoring based on WiFi as an example, a low-power communication strategy for terminal sensors is illustrated in Figure 1. The camera acts as a wireless active sensor connected to a 220 V power supply and communicates with the wireless gateway. Temperature, humidity, water level, and other sensors installed on the pipe wall or cable are passive sensing nodes that send sensing data to the wireless gateway when the energy meets the conditions. To avoid co-frequency interference between the energy harvesting and communication of the passive sensor node, the communication process is divided into three stages: the RF energy collection stage, the passive data backscatter or RF energy collection stage, and the active data transmission stage. The three stages operate in a cyclic manner. To avoid the confliction of the data transmission from passive sensor nodes to gateways, a distance–energy level coupling analysis method is proposed in the RF energy harvesting stage, and a wake-up delay correlation analysis method is proposed in the active data transmission stage to distinguish the timing of data transmission from each terminal sensor node.

## 3. System Model

### 3.1. Model Architecture

The cognitive radio network includes primary users and cognitive users. The primary user is the owner of the licensed channel and has priority in using the spectrum. Cognitive users do not have priority and can only occupy the licensed channel for information transmission when it is idle [28]. In the cognitive radio network, the primary user corresponds to the RF source in the system model of this article, and the cognitive user corresponds to the passive sensing node. The specific system model is shown in Figure 2.

In Figure 2, ➀, ➁, and ➂ represent, respectively, the different stages of the low-power communication strategy.

The system model consists of an RF source, a receiving gateway, and N passive sensing nodes. RF source refers to active terminal sensors that generally need to transmit large data packets of images and videos, such as infrared thermal imagers and cameras in power systems. Passive sensing nodes refer to passive terminal sensors that generally require the transmission of small amounts of data such as strings, temperature and humidity sensors, water level sensors, displacement sensors, etc. The wireless gateway is an active device that transmits sensing data from terminal sensors to access node devices connected to the detection platform. Considering the diverse types of terminal sensors for power system status monitoring, this article comprehensively considers the collaborative work of sensors for transmitting large and small amounts of data.

### 3.2. Mode of Operation

The previous section introduces the static description of a low-power communication system model based on cognitive backscatter. This section further introduces the dynamic working logic of this system model. In order to ensure that the communication and energy acquisition use the same frequency band and do not interfere with each other, the following settings are made:

Setting 3. The RF source is synchronized with the built-in clock of the wireless gateway. Taking Figure 3 as an example, start from time $t_{1}$ and reach $t_{2}$ and $t_{3}$ at the same time.

Setting 4. This strategy divides the low-power communication system model based on cognitive backscatter into three stages, namely the passive sensing node RF energy harvesting and information transmission stage (hereinafter referred to as the “first stage”), the RF source information transmission stage (hereinafter referred to as the “second stage”), and the passive sensing node information transmission stage (hereinafter referred to as the “third stage”).

Energy harvesting and information transmission of passive sensing node RF stage

At this stage, the RF source emits carrier waves without information to charge each passive sensing node, which is in a busy state. After receiving the carrier signal from the wireless gateway at time $t_{1}$, each passive sensing node determines the data transmission time based on the distance–energy level coupling analysis method. If it reaches its control energy level, the data collected using the sensor unit are modulated and transmitted to the wireless gateway through backscatter communication. At this stage, each passive sensing node can only send data no more than once, and the rest of the time, each one is used for RF energy collection to avoid collisions between multiple passive sensing nodes sending data. It should be noted that passive sensing nodes will only send data when they receive broadcast signals. Passive sensing nodes that cannot reach the control level in this stage will not send data again even if they reach the control level after receiving the next stage message to avoid occupying the channel.

RF source information transmission stage

At this stage, the RF source sends data to the wireless gateway and is in a busy state. The header of the message sent by the RF source carries a time series, which is received by the wireless gateway for time calibration. Once the passive sensing node decodes a message with time information, it stops reading the information and only collects RF energy. If certain passive sensing nodes have insufficient energy and cannot receive RF source information at the beginning of this stage, they will send sensing data to the wireless gateway when the energy reaches the control level. To avoid this situation, the RF source will send multiple messages carrying time series at the beginning of this stage, and then only send valid information thereafter.

Passive sensor node information transmission stage

At this point, the wireless gateway detects that the RF source is in an idle state and sends a broadcast signal to each passive sensing node. Each passive sensing node first determines the time to send data based on the wake-up delay correlation analysis method. To ensure the probability of correct data transmission, all passive sensing node adopts the sending method of doing everything possible. Therefore, in this stage, each passive sensing node can only send data once at most.

## 4. Key Technology

### 4.1. RF Energy Harvesting Technology

Collecting the RF energy generated during the communication process of wireless signal sources is a key technology for energy harvesting of passive sensors in complex working scenarios [29]. Its technical principle can be summarized as follows: The RF energy is converted into AC power and input into the matching network. The RF antenna matching network then transmits the AC power to the rectifier to convert it into DC power and store it to drive passive sensors for extremely low-power data transmission. The energy harvesting receiver (corresponding to the passive sensing node in this article) directly rectifies the RF signal and obtains the DC current to charge the energy storage component. Therefore, this technology can provide a theoretical basis for the passive operation of small data terminal sensor nodes in status monitoring, and has been widely applied in scenarios such as passive IoT and passive NFC intelligent locks. Based on this, this article introduces this technology into the scenario of PPGs, expecting to provide continuous and reliable energy supply for large-scale small-data terminal sensors by combining it with cognitive backscatter communication technology.

For the convenience of technical discussions in subsequent chapters, the working principle of RF energy harvesting technology is briefly described below:

The process of RF energy harvesting is shown in Figure 4, where the RF source is in the stage of sending information, and the passive sensing nodes are in the state of receiving energy. The mathematical expression of the signal $R\left(t\right)$ received by the passive sensing node is shown in Equation (1):(1)$$R\left(t\right)=\frac{\lambda \sqrt{P_{s}G_{s}G_{r}}hs\left(t\right)}{\sqrt{{\left(4\pi d\right)}^{k}}}+n_{s}\left(t\right)$$

In Equation (1), h and d is the channel gain and distance from the RF source to the passive sensing node, respectively. Channel gain characterizes the effect of the channel on the signal. At the transmitting end, the signal is sent with a certain power and amplitude. During the transmission through the channel, it will be affected by various factors, such as multipath effect, etc. Multipath effect causes the signals to superimpose at the receiving end, which may lead to signal enhancement or attenuation, thus causing fluctuations in the channel gain. The magnitude of the gain depends on the phase relationship and amplitude of each multipath signal, ultimately affecting the magnitude of the received power; λ is the operating wavelength; $P_{s}$ is the signal transmission power of the RF source; $G_{s}$ is the gain of the transmitting antenna; $G_{r}$ is the gain of the receiving antenna; k is the path loss exponent [28], and, in free space, the value of k is taken as 2; and $s\left(t\right)$ is the normalized signal emitted by the RF source, as shown in Equation (2):(2)$$E\left\{{\left|s\left(t\right)\right|}^{2}\right\}=1$$

The energy collected by passive sensing nodes is mathematically related to the transmission distance and energy collection duration, as shown in Equation (4):(3)$$P=\frac{\mu \lambda^{2}P_{s}G_{s}G_{r}{\left|h\right|}^{2}}{{\left(4\pi d\right)}^{k}}$$

In Equation (3), P is the power collected by the passive sensing node and μ is the energy conversion efficiency of the passive sensing node.(4)E=P⋅T

In Equation (4), E is the power and energy collected by the passive sensing node and T is the duration of energy collection.

### 4.2. Methodology for Avoiding Data Conflicts

Distance energy level coupling analysis method

Considering that due to the limitation of power consumption for passive sensing nodes in low-power communication systems, it is impossible to perform carrier sensing and clock synchronization. When multiple passive sensing nodes send data simultaneously, collisions are inevitable, resulting in data transmission failures and consuming the electricity accumulated in the energy storage components during the RF energy collection stage, which undermines the rationality of the theoretical establishment of low-power communication systems. Therefore, this article proposes a distance–energy level coupling analysis method to ensure smooth communication between each passive sensing node and the gateway during the RF energy acquisition and information transmission stage.

In this method, “distance” refers to the distance between the passive sensing node and the RF source when collecting RF energy; “energy level” refers to the control energy level that a passive sensing node needs to be able to send data. Energy level quantifies the energy stored in the energy storage components of the passive sensing node, as shown in (4):(5)$$E_{i}=\frac{E}{\delta }$$

In Equation (5), $E_{i}$ represents the energy level of the passive sensing node i; δ represents the quantified energy value, and classifies the energy collected by the passive sensing node.

The specific method for distance–energy level coupling analysis is as follows:

Step 1: Determine the emission control energy level corresponding to each passive sensing node; assuming that the distance from the passive sensing node to the RF source is $d_{i}$, the linear distance values from all passive sensing nodes in the system to the RF source can form an array $d=\left[d_{1},d_{2},\ldots ,d_{N}\right]$, N is the total number of passive sensing nodes in the low-power communication system. Sort the distance values between each passive sensing node and the RF source from small to large to obtain a sequence array $d^{\prime }=\left[d_{5},d_{N},\ldots ,d_{N-6}\right]$. For the convenience of discussions, the sorted array $d^{\prime }$ is equivalent to $D=[D_{1},D_{2},\ldots ,D_{N}]$.

Step 2: Determine the control energy levels for each passive sensing node based on the distance array between the passive sensing nodes and the RF source. Using Equation (6) to obtain the control energy levels corresponding to two adjacent passive sensing nodes in the distance array D,(6)$$W_{i+1}=\left\{\begin{array}{ll} W_{i}+a, & D_{i+1}-D_{i}<L\\ W_{i}, & D_{i+1}-D_{i}\geq L \end{array}\right.$$

In Equation (6), $D_{i}$ is the distance between passive sensing node i and the RF source in the sorted distance array, L is the distance threshold, $W_{i}$ is the control energy level corresponding to passive sensing node i, and a is the energy level adjustment parameter.

This step can avoid collisions between the data sent at the same time when the energy collection efficiency are close to each due to the similarity in distance values between some passive sensing nodes and the RF source.

Step 3: Each passive sensing node determines whether to send data or to accumulate energy at that moment by comparing the magnitude relationship between the energy level and the control energy level. If the energy level is greater than the control level, data will be sent; otherwise, energy will be accumulated.

Formula (5) considers that the charging speed of passive sensing nodes in low-power communication systems is greatly affected by the energy transmission distance: the closer the passive sensing node is to the RF source, the greater the energy storage per unit time. Therefore, if the distance difference between adjacent passive sensing nodes in the array D is greater than L, it indicates that the time difference between the two passive sensing nodes to reach the same energy level is large, and the control energy levels of these two passive sensing nodes can be set at a same level. If the distance difference between adjacent passive sensing nodes in the array D is less than L, it indicates that the time difference between these for the two passive sensing nodes to reach the same energy level is small, and their control energy levels needs to be distinguished. Using this method, the time when the energy accumulation of passive sensing nodes reaches the control level is divided based on the charging efficiency. The terminal sensors of PPGs are mainly used for small data volume services, such as temperature, humidity, water level, and other sensors. Therefore, after the gateway sends the broadcast signal to obtain the data, as long as it distinguishes the time of the data sent between each passive sensing node, the collision probability can be reduced.

In the method of this paper, for each additional node, the energy level of each node needs to be calculated separately at each quantified moment in the simulation. Therefore, the computational complexity of this method is $O\left(n\right)$.

Wake up-delay correlation analysis method

In the third stage, the battery levels in the energy storage components are different after the work in the previous two stages. However, it still needs to maintain a low collision probability. A wake up-delay correlation analysis method is proposed to address this issue. The specific method is as follows:

Step 1: Set the delay duration for each passive sensing node. The further away from the RF source, the poorer the ability to accumulate power, and that passive sensing nodes require energy consumption to delay sending timing. Consequently, the corresponding delay duration array $D=[D_{1},D_{2},\ldots ,D_{N}]$ for the rearranged distance values of each passive sensing node is shown in (7).(7)$$Τ=[\left(N-1\right)\tau ,\left(N-2\right)\tau ,\ldots ,\tau ,0]$$

In Equation (7), τ is the duration of delayed data transmission between two adjacent passive sensing nodes in D. According to the order from far to near, each node sends data at intervals of τ. The research object in this article is the small-data passive terminal sensor; with small data volumes, collisions can be avoided by distinguishing the sending time of each passive sensor.

Step 2: When the receiving gateway confirms that the RF source is idle, it sends a broadcast message to the passive sensing node, informing it that the collected data can be sent.

Step 3: After receiving the broadcast signal, each passive sensing node determines the delay duration by array Τ. There are two types of states for all passive sensing nodes: (1) after the timing is completed, if there is still battery remaining, the data will be sent to the receiving gateway “as much as possible” according to the principle of backscatter communication. After this action is completed, all of the battery power of the energy storage component will be consumed; (2) if the timing causes the internal battery of the terminal sensor node to run out, no data will be sent before the next round.

After the third stage, the power of all passive sensing nodes is zero, ensuring that each passive sensing node in the first stage of the next round completes data transmission smoothly based on the distance–energy level coupling analysis method. The gateway waits for time $\beta_{3}$ to enter the first stage of the next round after sending out the broadcast signal. The waiting time satisfies are as shown in (8):(8)$$\beta_{3}=N\tau$$

## 5. Case Study

### 5.1. Overview of the Simulation Environment

In order to verify the effectiveness of the method in this paper, a simulation environment for the underground transmission gallery is built. A series of passive sensor nodes, a camera (active sensor), and a wireless gateway are deployed according to the system model in Section 3.1. Among them, the passive sensor nodes are all sensors for small data volume services, which have the function of measuring environmental parameters such as humidity, water level, gas concentration, etc. The simulation environment is shown in Figure 5.

As shown in Figure 5, the characteristics of the simulation environment are described below:
The underground pipe tunnel has a diameter of 10 m and a length of 20 m with low temperature, high humidity, low visibility, and no continuous light source for most of the time.Based on known business experience and historical monitoring data of power equipment, a series of observation points were selected to deploy passive sensing nodes, which were, respectively, deployed on pipe walls, cable holders, cable branch boxes, and other locations, as shown in Figure 5 and Table 1. When selecting the location of the passive sensing nodes, the position of the camera (RF source) (10, 0, 5) is used as the reference coordinate, and the maximum linear distance between the passive sensing node and the reference coordinate does not exceed 14 m to ensure that the passive sensing node can receive RF signals.

According to Table 1, based on the historical frequency of fault alarms at each location of the power equipment in the PPGs, ten observation positions were set and there are four observation levels; one passive sensing node was deployed at each position.

### 5.2. Experimental Procedure

#### 5.2.1. Comparative Experiment of Method Performance

In this subsection, the Pure ALOHA method is selected as the object for horizontal comparison. The Pure ALOHA algorithm is a random-access channel access protocol. Its core lies in that nodes can send data at any time without the need to make reservations or wait for specific moments. It is completely based on the random principle. When a node has data to transmit, it immediately sends the data to the channel. After sending, it determines whether a conflict has occurred through the feedback from the receiving party or self-detection. If a conflict occurs or no confirmation information is received, it will randomly wait for a period of time and then re-transmit [30]. The relevance between this method and the proposed method is that they are both communication methods that do not require an internal clock and are suitable for low-power communication scenarios.

The Slotted ALOHA algorithm is a random-access protocol improved on the basis of the Pure ALOHA algorithm. It divides time into equal-length time slots and requires nodes to send data only at the beginning of the time slots, thus making the data transmission discrete in time. Nodes need to cache data first and wait for appropriate time slots. After sending, they also determine whether to re-transmit according to the confirmation information. This algorithm greatly reduces the probability of conflicts and improves the channel utilization rate through the time-slot synchronization mechanism. However, it requires time synchronization among nodes, which makes relatively high demands on the system and is not suitable for low-power-consumption sensing scenarios [31].

Contrast experiments are carried out in terms of three aspects, the system communication collision probability, the amount of data successfully transmitted by the system, and the system communication energy consumption, to analyze the performance differences between the two under the condition of the increase in passive sensing nodes. The specific experimental parameter settings include the following: The matrix of the number of passive sensing nodes N is [2, 3, 4, 5, 6, 7, 8, 9, 10]. The maximum value of the control energy level is 130. The power difference between adjacent energy levels is −10 dBm. The maximum distance between the passive sensing node and the camera is 14 m. The channel gain h follows a gamma distribution where the parameter b=0.25. Other specific experimental parameters set in the simulation are shown in Table 2.

It should be noted that, in the simulation, parameters such as camera transmission power, passive sensor node energy conversion efficiency, path loss index, and channel gain in the ALOHA protocol are consistent with the method parameters proposed in this paper, and the control energy level of each passive sensor node is set to 3.

In the comparative experiment, the first and second passive sensing nodes with the highest observation level were set as the initial observation nodes, and subsequent observation nodes were added in descending order of observation level according to Table 1. Compared to the data access method of ALOHA, the data access method proposed in this paper can set differentiated control levels for different passive sensing nodes in order to achieve multiple access for a series of passive sensing nodes.

In this simulation experiment, the interval collecting data on the status of PPGs is set to 15 min. During the simulation process, 3000 rounds were conducted on nine different passive sensing node distribution patterns, each round lasting 5 min. Three rounds were grouped together, and the same collected data were sent. If the data transmission was successful in one or more rounds within a group, the system’s data transmission was considered successful. Otherwise, it was considered a failure. To improve the difference in data transmission time between each passive sensing node in each round of each group, and to avoid repeated data transmission failures of individual nodes, the distance thresholds for the three rounds were set at 3 m, 4 m, and 5 m, respectively, to ensure monitoring effectiveness. The message length for a passive sensing node to send observation data once is 2 bytes, and the simulation results are shown in Figure 6.

#### 5.2.2. Experimental Study on the Influence of Parameters on Performance of the Method

As can be seen in Section 4.2, the energy level adjustment parameter *a* and the distance threshold L jointly affect the timing of data transmission by passive sensing nodes. Therefore, there exists a certain degree of causal relationship between these two parameters and the probability of system communication collision. This section will conduct an in-depth analysis of the causal relationship between the above two parameters and the probability of system communication collision, as detailed below:If the energy level modulation parameter value is small, the two passive sensor nodes with similar distance values from the camera need to satisfy similar control energy levels, and vice versa.If the distance threshold value L is small, the control energy levels of the two passive sensor nodes that are close to the camera need to be different, and the control energy levels of the passive sensor nodes can be spread out. In contrast, the control energy levels of the passive sensor nodes are close to each other.In the simulation experiments in this section, the value range in the distance threshold L is set to [1, 10], and the value range in the energy level adjustable parameter is set to [1, 9]. The experimental results of the impact of the distance threshold L and the adjustable parameter a on the system communication collision probability are shown in Figure 7.


In Figure 7, the minimum probability of system communication collision is marked in red, with coordinates of (3, 9, 1.23), which means the distance threshold L is 3 m and the energy level adjustment parameter a is 9. The corresponding system communication collision probability is 1.23%.

### 5.3. Experimental Results

According to the above experimental process, the performance advantages of this method compared with the ALOHA method are analyzed. In order to quantitatively compare the performance of the two methods, three evaluation indicators are given below. The calculation method is given as Equations (9)–(11).(9)$$\Delta p=p_{c}-p_{f}$$

In Equation (9), Δp is the reduction in collision probability of the system model in this method compared to the ALOHA method; $p_{c}$ and $p_{f}$ is the system communication collision probability array corresponding to the number of passive sensor nodes in the process of low-power communication according, respectively, to the ALOHA method and this method.(10)$$\Delta y=\frac{y_{f}-y_{c}}{y_{c}}\times 100\%$$

In Equation (10), Δy refers to the increment in the data which is successfully sent by the system model; $y_{c}$ and $y_{f}$ are the array of data successfully sent by the system corresponding to the number of passive sensor nodes in the process of low-power communication based on the ALOHA method and this method, respectively.(11)$$\Delta e=\frac{e_{c}-e_{f}}{e_{c}}\times 100\%$$

In Equation (11), Δe is the energy consumption reduction in the system model; $e_{c}$ and $e_{c}$ are the system communication energy consumption array corresponding to the number of passive sensor nodes in the process of low-power communication based on the ALOHA method and this method, respectively.

It can be seen that the above three formulas can systematically evaluate the performance difference between the method in this paper and the ALOHA method in low-power wireless communication systems.

Specifically, Formula (9) represents the performance at the level of communication collision probability, Formula (10) represents the performance in successfully sending data volume in unit time, and Formula (11) evaluates the difference in system communication energy consumption under the condition that the two methods complete the same communication task.

The performance difference between the two methods is calculated according to Equations (9)–(11), as shown in Table 3.

#### 5.3.1. Conclusion of Method Performance Comparison Experiment

The following conclusions could be summarized by the above experiment process and data.

(1)It can be seen from Figure 6a and Table 3 that, with the increase in the number of passive sensor nodes, the communication collision probability of this method and the ALOHA method gradually increases, and the increase in this method is significantly less than that of the ALOHA method. When the number of passive sensor nodes reaches or exceeds 7, the communication collision probability difference between the two methods exceeds 5%. The above experimental results show that, on the one hand, with the increase in the number of passive sensor nodes, the number of communication messages of the system model also increases, which objectively increases the probability of communication collision; on the other hand, the method in this paper adopts the distance–energy level coupling analysis method and the wakeup-delay correlation analysis method to avoid conflicts in the low-power communication process of the terminal sensors of the condition monitoring of the underground transmission tunnel, which effectively reduces the communication collision probability to a certain extent. However, the ALOHA method has no conflict avoidance mechanism, so its communication collision probability is high. Therefore, this method has better performance than the ALOHA method in terms of communication collision probability.(2)It can be seen from Figure 6b and Table 3 that, with the increase in the number of passive sensor nodes, the amount of data successfully sent by both methods shows a trend of downward. Further analysis shows that the amount of data successfully sent by this method increases with the increase in the number of passive sensor nodes, compared to the ALOHA method. When the number of passive sensor nodes reaches or exceeds 9, the difference in the amount of data successfully is more than 10% of the ALOHA method. The above experimental results show that, on the one hand, with the increase in the number of passive sensor nodes, the communication collision probability of the system model increases, which objectively reduces the proportion of successful message transmission in unit time; on the other hand, the method in this paper improves the probability of successful data transmission in three rounds and reduces data loss, while the ALOHA method has no conflict avoidance mechanism. With the increase in the number of passive sensor nodes, the data loss is aggravated. Therefore, in terms of the amount of data successfully sent, the method in this paper performs better than the ALOHA method.(3)It can be seen from Figure 6c and Table 3 that, with the increase in the number of passive sensor nodes, the communication energy consumption of the system model of the two methods increases gradually, and the increase in the communication energy consumption of the system model of this method is significantly smaller than that of the ALOHA method. When the number of passive sensor nodes exceeds 5, the difference between the communication energy consumption of the system model of the two methods increases significantly. The reason for this is that, with the increase in the number of passive sensor nodes, the communication collision probability of data transmission of the system increases subsequently, which objectively increases the communication energy consumption of the system model under the condition of completing the same communication task. Therefore, compared to the ALOHA method, this method has better performance in system communication energy consumption.

In conclusion, based on the comparative experiments in Figure 6 and Table 2, this method has performance advantages over traditional methods in terms of communication collision probability, data transmission volume, and communication energy consumption due to the combination of RF energy collection and cognitive backscatter technology.

#### 5.3.2. Conclusions of Experiment on Parameter Influence

Analysis of the effect of energy level adjustment parameter a on performance

It can be seen from Figure 7a that the collision probability of the system shows an obvious downward trend with the increase in the energy level adjustment parameter a. The reason for this is that large energy level adjustment parameters can clearly distinguish the control energy level of each passive sensor node, so that each passive sensor node has a differentiated data transmission time, reducing the probability of communication collision. Therefore, based on the simulation process and results, the relationship between the energy level adjustment parameter a and the communication collision probability is analyzed. The communication collision probability of the system is the lowest when the value of parameter a is 9 (the maximum value of a).

2.Analysis of the influence of distance threshold

It can be seen from Figure 7b that when the distance threshold L reaches 3 m, the collision probability of system communication decreases significantly and then tends to stabilize. The reason for this can be summarized as follows: according to Equation (5), the distance threshold L affects the control energy level of each passive sensor node, and the control energy level further determines the time when each passive sensor node sends data, which directly affects the system communication collision probability. For the ten node coordinates in Table 1 of this example, we can obtain D = [4 7 8 8 9 10 10 11 11 13] and then calculate the difference in the distances in *D* to form an array $D^{\prime }$ = [3, 1, 0, 1, 1, 0, 1, 1, 2]. In set $D^{\prime }$, except that the distance difference between the second node and the first node is 3, the distance difference between the other nodes is in [0, 2]. Therefore, when the distance threshold L is set at [0, 2], the control level of each passive sensor node will be greatly affected. The simulation results shown in Figure 8 further reveal the correlation between the distance threshold L and the control energy level.

Figure 8 can be described as follows: when the energy level adjustment parameter a is set to 9, the distance threshold L corresponds to the control energy levels of each passive sensor node in the interval [1, 10].

The following can be seen from Figure 8: (1) When the distance threshold L is 1 m, the control energy levels of the 1st, 2nd, 9th, and 10th passive sensing nodes are the same. (2) When the distance threshold L is 2 m, only the control energy levels of the 1st and 2nd passive sensing nodes are the same. (3) When the distance threshold L is greater than or equal to 3 m, the control energy levels of each passive sensor node increase incrementally. There is no situation where the control energy levels are the same, which improves the difference in the control energy levels of each passive sensing node. Moreover, the control energy levels of each passive sensing node no longer change with the variation in the distance threshold L.

#### 5.3.3. Limitations and Prospects

The simulation experiment demonstrates that the low-power communication strategy based on cognitive backscatter proposed in this paper can reduce the system communication collision probability to a certain extent, but when the number of passive sensor nodes reaches 6, the system communication collision probability exceeds 7%. Therefore, in the future, we can consider optimizing the low-power communication strategy to achieve lower system communication collision probability.

The PPG is mostly cylindrical in shape parallel to the ground, but there are also cubes parallel to the ground or shapes that need to bend and turn. Therefore, future research should focus on the deployment of passive sensor nodes and the optimization of their communication performance within a complex three-dimensional space.

## 6. Conclusions

The number of sensing nodes used for state monitoring at the terminal of power systems is increasing, and it is difficult to use active power supply, so the laying of communication cables is increasingly complex. To solve this problem, this paper proposes a low-power communication strategy based on cognitive backscattering. The conclusions of this paper are as follows:In view of the fact that the online monitoring devices of PPGs need to be operated in an active manner, which is difficult to deploy in scale, and that the current business needs of cable equipment and cable channel monitoring and communication cable laying are complex, a technical framework for low-power communication of terminal sensors is proposed.A low-power communication system model of a terminal sensor is established, and its dynamic working process is divided into three stages to ensure that the same frequency band of communication and energy acquisition do not interfere with each other during the system’s operation.In the low-power communication strategy, the corresponding key technologies, such as RF energy acquisition, range level coupling, and the wake-up delay correlation analysis method, are proposed to reduce the collision probability of low-power terminal sensor data transmission.Through simulation experiments, the performance advantages of this paper compared to the traditional communication mode in the three aspects of system communication collision probability, successful data transmission, and system communication energy consumption are demonstrated. By studying the correlation between the energy level adjustment parameter a and the distance threshold L on the system communication collision probability, we can further optimize the following parameters: when the distance threshold L is 3 m, and when the energy level adjustment parameter a is 9, the minimum communication collision probability of the system is 1.23%.

In the future, studies could consider integrating the low-power communication strategy of cognitive backscattering the terminal sensors to carry out field experiments in underground transmission tunnels.

## Figures and Tables

**Figure 1 sensors-25-01317-f001:**
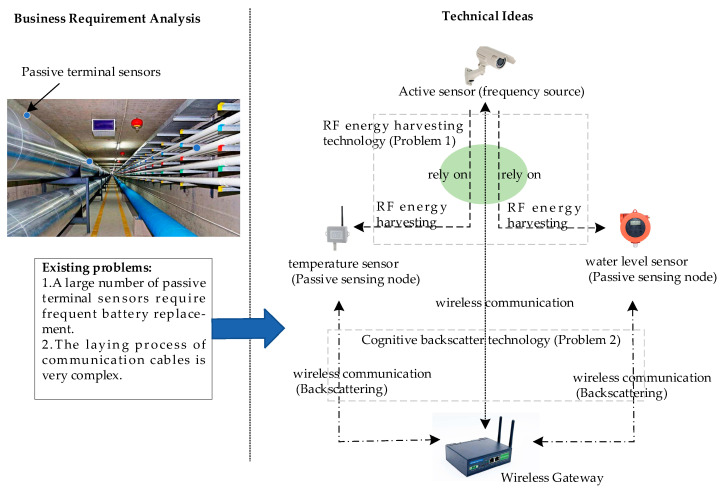
Low-power communication technology framework for terminal sensors.

**Figure 2 sensors-25-01317-f002:**
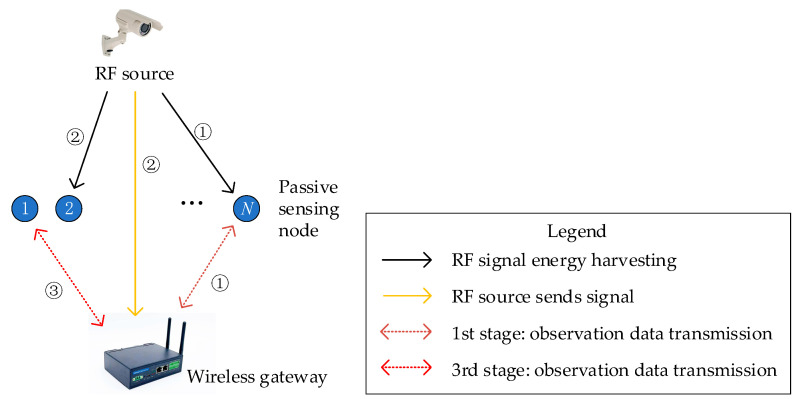
Low-power communication system model based on cognitive backscatter.

**Figure 3 sensors-25-01317-f003:**
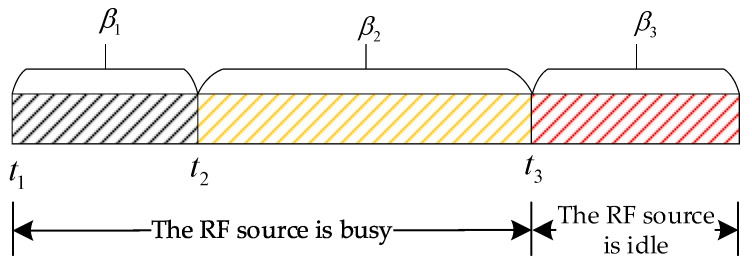
Time distribution of each stage in one round.

**Figure 4 sensors-25-01317-f004:**
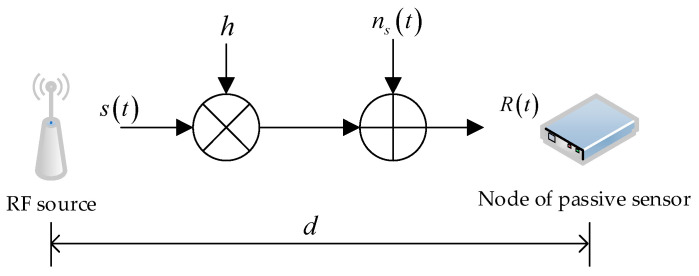
Schematic diagram of radio frequency energy harvesting process.

**Figure 5 sensors-25-01317-f005:**
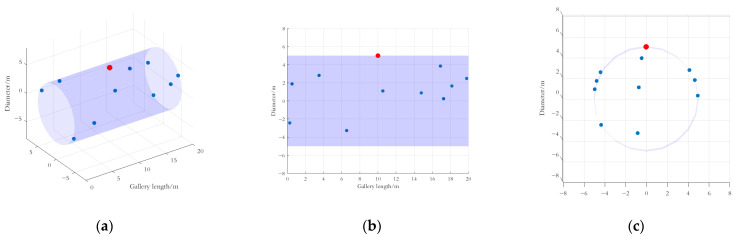
Simulation environment of PPGs. (**a**) Stereoscopic diagram of simulation environment. (**b**) Front view of simulation environment. (**c**) Side view of simulation environment. In this figure, the positions of the cameras are marked in red, and the deployment locations of sensors for various small data volume services are marked in blue.

**Figure 6 sensors-25-01317-f006:**
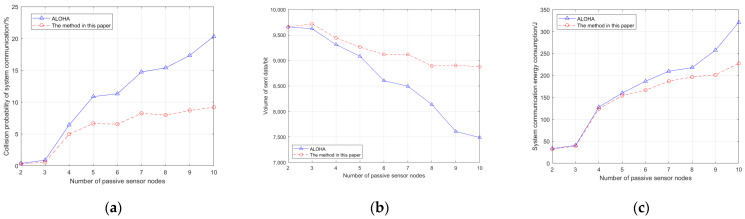
Performance comparison of the methods in the comparative experiments. (**a**) Comparison of the collision probability performance in system communication. (**b**) Performance comparison of the sent data volume in the system. (**c**) Comparison of system communication energy consumption performance.

**Figure 7 sensors-25-01317-f007:**
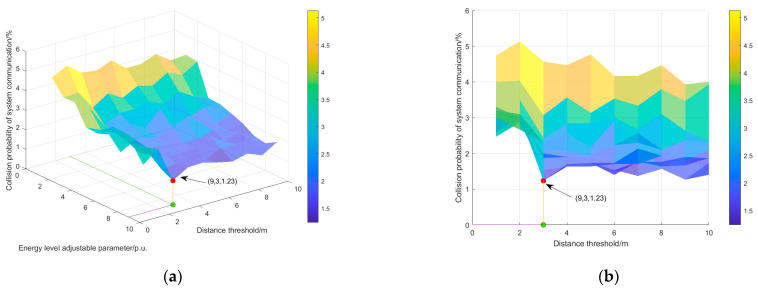
The relationship between the parameters and the communication collision probability of the system. (**a**) Stereoscopic diagram of the relationship between the parameters and the probability of system communication collision. (**b**) Front view of the relationship between the parameters and the probability of the system communication collision.

**Figure 8 sensors-25-01317-f008:**
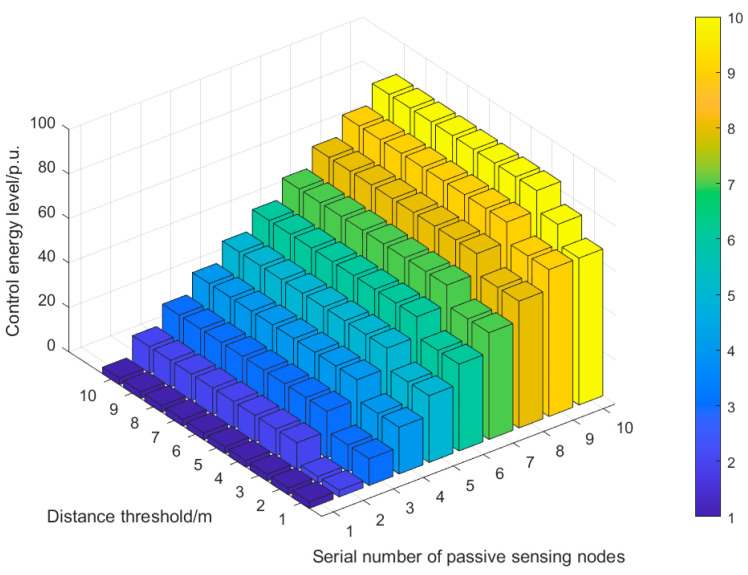
Relationship diagram between distance threshold and control energy level of each passive sensor node.

**Table 1 sensors-25-01317-t001:** Types and observation positions of passive sensor nodes.

No.	Coordinate Position	Number of Historical Alarm	Observation Level	Linear Distance from RF Source/m
1	(19.79, −4.34, 2.48)	11	High	11
2	(14.78, −4.92, 0.89)	11	High	8
3	(18.16, −4.72, 1.66)	8	Medium high	10
4	(10.54, −0.70, 1.10)	8	Medium high	4
5	(16.89, −0.40, 3.85)	4	Medium	7
6	(0.26, −4.37, −2.43)	3	Medium	13
7	(6.55, −0.83, −3.27)	2	Medium	9
8	(0.52, 4.63, 1.88)	0	Low	11
9	(17.25, 4.99, 0.25)	0	Low	10
10	(3.50, 4.13, 2.82)	0	Low	8

**Table 2 sensors-25-01317-t002:** Specific experimental parameters in the simulation.

Experimental Parameters	Value
$P_{s}$	14 dBm
μ	0.5
λ	0.588 m
$G_{s}$	6 dB
$G_{r}$	6 dB
$\beta_{1}$	90 s
$\beta_{2}$	120 s
$\beta_{3}$	90 s
k	2
L	3
$W_{1}$	3
a	2
b	0.25

**Table 3 sensors-25-01317-t003:** Performance comparison of two methods.

*N*	Δp	Δy/%	Δe/%
2	0.06	0.00	4.33
3	0.33	1.00	2.96
4	1.45	1.42	3.13
5	4.23	1.98	3.75
6	4.78	6.00	10.70
7	6.5	7.34	10.95
8	7.43	9.29	9.63
9	8.63	17.03	22.09
10	11.16	18.59	29.28

## Data Availability

The data presented in this study are available on request from the corresponding author due to participant privacy restrictions.

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
