# Peer review of "A Low-Power Communication Strategy for Terminal Sensors in Power Status Monitoring"

_sensors, 2025, doi:10.3390/s25051317_

Round 1
Reviewer 1 Report
Comments and Suggestions for Authors
The paper presents a framework for simulation for low-power communication of terminal sensors based on RF energy harvesting and cognitive backscatters to avoid communications collisions.
However, some points should be addressed:
11. Parameter Table: Include a table summarizing all parameters used in the simulations, along with their descriptions and values. This will provide clarity and make it easier for readers to understand the framework.
22. Transmitted Power: Adjust the transmitted power to realistic values. For Wi-Fi, the maximum allowed power is 20 dBm (EIRP) for 2.4 GHz and up to 36 dBm for some 5 GHz channels and countries (in other is limited to 20 dBm). Simulations should be done with these powers.
33. Received Power: Include the received power at the positions of the passive nodes. This will help in understanding the power distribution and its impact on the network.
44. Minimum Power for Charging: Specify the minimum power required to start charging the batteries. This is crucial as DC-DC converters need a minimum input voltage to operate effectively.
55. Passive Mode Sensors: Clarify whether the sensors in passive mode require a battery (semi-passive tags) or are completely passive, relying on RF signals for energy. Indicate the threshold power needed for operation and include it in the model.
66. Channel Gain Parameter: Rewrite equation (3) using the Friis transmission equation, which accounts for the transmitter and receiver antenna gains and the frequency. Provide references for the path loss index (k) value used, especially for indoor scenarios.
77. ALOHA Algorithm: Provide details about the ALOHA algorithm, including its variants like pure ALOHA and slotted ALOHA. Include references to support the explanation.
88. Multipath Effects: Discuss the impact of multipath propagation on received power. Multipath can cause variations in received power due to constructive and destructive interference.
By addressing these points, the framework will be more comprehensive and realistic.
Author Response
Comments 1: Parameter Table: Include a table summarizing all parameters used in the simulations, along with their descriptions and values. This will provide clarity and make it easier for readers to understand the framework.
Response 1: Thank you for pointing this out. We agree with this comment.In the paper, the parameters used in the simulation and their values are presented in tabular form. Please refer to Table 2 in the re - uploaded manuscript. The location of the revision is on line 413 of the new manuscript.
Comments 2: Transmitted Power: Adjust the transmitted power to realistic values. For Wi-Fi, the maximum allowed power is 20 dBm (EIRP) for 2.4 GHz and up to 36 dBm for some 5 GHz channels and countries (in other is limited to 20 dBm). Simulations should be done with these powers.
Response 2: Thank you for pointing this out. We agree with this comment. In the PPGs scenarios, the cameras communicate via Wi - Fi. Therefore, we have revised the data and set the transmit power to 20 dBm. We have re - simulated the examples, and Figures 6 - 8 in the fourth part of the new manuscript have all been revised. The location of the revision is in Chapter 4 of the new manuscript.
Comments 3: Received Power: Include the received power at the positions of the passive nodes. This will help in understanding the power distribution and its impact on the network.
Response 3: Thank you for pointing this out. We agree with this comment. We have added Equation 3 as the received power of the passive sensing node, and Equation 4 represents the corresponding energy. The location of the revision is on line 258 of the new manuscript.
Comments 4: Minimum Power for Charging: Specify the minimum power required to start charging the batteries. This is crucial as DC-DC converters need a minimum input voltage to operate effectively.
Response 4: Thank you for pointing this out. We agree with this comment. Based on relevant literature references and our team's previous research, the minimum power required to charge a passive sensing node is -10 dBm.
Comments 5: Passive Mode Sensors: Clarify whether the sensors in passive mode require a battery (semi-passive tags) or are completely passive, relying on RF signals for energy. Indicate the threshold power needed for operation and include it in the model.
Response 5: Thank you for pointing this out. We agree with this comment. In our research, the passive sensing nodes are pure passive sensors that obtain energy solely from radio - frequency signals. The threshold power required for the operation of the sensor is -10 dBm.
Comments 6: Channel Gain Parameter: Rewrite equation (3) using the Friis transmission equation, which accounts for the transmitter and receiver antenna gains and the frequency. Provide references for the path loss index (k) value used, especially for indoor scenarios.
Response 6: Thank you for pointing this out. We agree with this comment. Formula 3 has been rewritten according to the Friis transmission formula, and Formula 1 and Formula 4 have been modified accordingly. k is the path - loss exponent. In free space, the value of k is taken as 2. The reference for the value of k is [28], which has been added on line 255 of the new manuscript.
Comments 7: ALOHA Algorithm: Provide details about the ALOHA algorithm, including its variants like pure ALOHA and slotted ALOHA. Include references to support the explanation.
Response 7: Thank you for pointing this out. We agree with this comment. The new manuscript has supplemented the ALOHA algorithms, including the Pure ALOHA algorithm and the Slotted ALOHA algorithm. The references have been cited at the corresponding positions. The modified part is located in Subsection 4.2.1 of the new manuscript, corresponding to Line 384.
Comments 8: Multipath Effects: Discuss the impact of multipath propagation on received power. Multipath can cause variations in received power due to constructive and destructive interference.
Response 8: Thank you for pointing this out. We agree with this comment. Channel gain represents the effect of the channel on the signal. At the transmitter, the signal is emitted with a certain power and amplitude. During the transmission through the channel, it is subject to various factors, such as the multipath effect. The multipath effect causes the signals to superimpose at the receiver, which may lead to signal enhancement or attenuation, thus causing fluctuations in the channel gain. The magnitude of this gain depends on the phase relationships and amplitudes of the multipath signals, ultimately affecting the magnitude of the received power. The modification is located at line 249 of the new manuscript.
Point 1: The English is fine and does not require any improvement.
Response 1: Thank you for pointing this out. We agree with this comment. We have refined the English expressions in the manuscript.
Additional clarifications
During the process of modification based on the expert's opinions, through research, it was found that the energy harvesting effect of radio frequency is relatively ideal when the distance is within 20 meters. Therefore, in the example calculation part, the maximum distance between each passive sensing node and the radio frequency source was changed to 20 meters, and the modeling and simulation were carried out again.

Reviewer 2 Report
Comments and Suggestions for Authors
The following are the reviewer's comments regarding the manuscript.
1. The manuscript introduces the motivation behind the development of low-power communication strategies for terminal sensors in power pipe galleries (PPGs). It identifies key challenges in existing methods, such as battery replacement issues and energy inefficiency. However, the introduction could benefit from additional context on how the proposed solution compares to other advanced low-power communication methods, particularly in terms of innovation.
What needs to be done:
a. Add a brief comparison of alternative methods and frameworks in low-power IoT communication for better contextualization.
b. The second paragraph in the introduction is too long. Please revise into shorter concise paragraphs.
c. Please remove the use of "Reference" in the manuscript. Simply cite the reference number without including the word "Reference".
2. The research design is well-structured, offering a clear explanation of the proposed technical framework based on cognitive backscatter communication. While the framework and system model are adequately described, a comparison with existing frameworks or baseline systems in terms of computational overhead and real-world feasibility would strengthen the impact.
3. The methods section outlines the cognitive backscatter communication strategy and its integration with RF energy harvesting in sufficient detail. However, the following points need clarification:
Simulation Details: The simulation environment's description is helpful, but including a schematic or diagram summarizing the stages would aid understanding.
Parameter Justification: While some parameters (e.g., energy level adjustment, distance thresholds) are explored, a more detailed justification of their selection is necessary.
4. The manuscript demonstrates strong experimental results comparing the proposed method with ALOHA. Key performance improvements are highlighted, such as reduced collision probability, increased data transmission volume, and lower energy consumption. However, the following is necessary:
What should be done:
Data Presentation: The results would benefit from more visual aids, such as heatmaps or 3D plots, to illustrate parameter dependencies and performance gains.
Discussion Depth: While the results are promising, the authors should provide a deeper discussion on potential limitations, such as scalability for larger PPG systems or sensitivity to environmental noise.
5. The conclusions are robust, which summarizes the innovative aspects and contributions of the study. They align well with the experimental results but could benefit from explicit mention of future work, such as real-world deployment scenarios or integration into existing IoT systems.
6. The authors should conduct minor editorial revisions to improve readability and flow.
Author Response
Comments 1: The manuscript introduces the motivation behind the development of low-power communication strategies for terminal sensors in power pipe galleries (PPGs). It identifies key challenges in existing methods, such as battery replacement issues and energy inefficiency. However, the introduction could benefit from additional context on how the proposed solution compares to other advanced low-power communication methods, particularly in terms of innovation.
What needs to be done:
a.Add a brief comparison of alternative methods and frameworks in low-power IoT communication for better contextualization.
b.The second paragraph in the introduction is too long. Please revise into shorter concise paragraphs
c. Please remove the use of "Reference" in the manuscript. Simply cite the reference number without including the word "Reference".
Response 1: Thank you for pointing this out. We agree with this comment.
a.In the second paragraph of the Introduction, References [20 - 22] introduce the backscatter communication architectures in low - power Internet of Things (IoT) communications, including single - and dual - site, ambient, parasitic, symbiotic, and symbiotic backscatter communication architectures that integrate active and passive transmissions.
b.Reference [23] in the original manuscript has been deleted, and the content of References [18 - 23] has been refined and simplified.
c. In the new manuscript, the reference numbers are directly cited, and the use of the word "Reference" has been removed.
Comments 2: The research design is well-structured, offering a clear explanation of the proposed technical framework based on cognitive backscatter communication. While the framework and system model are adequately described, a comparison with existing frameworks or baseline systems in terms of computational overhead and real-world feasibility would strengthen the impact.
Response 2: Thank you for pointing this out. We agree with this comment. Through analysis, we found that in the method of this paper, for each additional node, the energy level of each node needs to be calculated separately at each quantized moment in the simulation. The computational cost of this method is O(n). In the example simulation of this paper, the computational cost of the ALOHA method is also O(n). With the same computational cost, as can be seen from Figure 6, the method proposed in this paper has better performance compared with the traditional ALOHA method.
Comments 3: The methods section outlines the cognitive backscatter communication strategy and its integration with RF energy harvesting in sufficient detail. However, the following points need clarification:
Simulation Details: The simulation environment's description is helpful, but including a schematic or diagram summarizing the stages would aid understanding.
Parameter Justification: While some parameters (e.g., energy level adjustment, distance thresholds) are explored, a more detailed justification of their selection is necessary.
Response 3: Thank you for pointing this out. We agree with this comment.
In Figure 2, â‘ , â‘¡, and â‘¢ represent different stages of the low - power communication strategy respectively. In Figure 3, different time instants correspond to the three stages of the strategy proposed in this paper. The simulation in the example part of this paper is built based on the system model in Chapter 2.
As can be seen from Equation 6, a and L directly affect the control energy level of each passive sensing node, indirectly influence the time of data transmission, and ultimately determine whether a data transmission collision will occur.
Comments 4: The manuscript demonstrates strong experimental results comparing the proposed method with ALOHA. Key performance improvements are highlighted, such as reduced collision probability, increased data transmission volume, and lower energy consumption. However, the following is necessary:
What should be done:
Data Presentation: The results would benefit from more visual aids, such as heatmaps or 3D plots, to illustrate parameter dependencies and performance gains.
Discussion Depth: While the results are promising, the authors should provide a deeper discussion on potential limitations, such as scalability for larger PPG systems or sensitivity to environmental noise.
Response 4: Thank you for pointing this out. We agree with this comment.
We have modified all the figures in Part 4. For details, please refer to the figures in the experimental section.
In Subsection 4.3.3, the following statement was added to respond to the expert's opinion: This paper discusses the situation where the passive sensing nodes are within 20 meters of the radio frequency source. When it comes to larger scale PPG systems, the effectiveness of the method proposed in this paper may be affected, which awaits further research in the future.
Comments 5: The conclusions are robust, which summarizes the innovative aspects and contributions of the study. They align well with the experimental results but could benefit from explicit mention of future work, such as real-world deployment scenarios or integration into existing IoT systems.
Response 5: Thank you for pointing this out. We agree with this comment.In the last paragraph of Part 5, the future research directions are described. That is, it is considered to integrate the low - power communication strategy using cognitive backscattering into the sensor terminals and conduct on - site experiments in underground power transmission corridors.
Comments 6: The authors should conduct minor editorial revisions to improve readability and flow.
Response 6: Thank you for pointing this out. We agree with this comment. We have made numerous revisions to the manuscript to enhance the readability and fluency of the paper. For instance, lines 259 and 261 in the new manuscript, among others.
Point 1: The English is fine and does not require any improvement.
Response 1: Thank you for pointing this out. We agree with this comment. We have refined the English expressions in the manuscript.

Reviewer 3 Report
Comments and Suggestions for Authors
The paper uses a low-power communication strategy for terminal sensors in power pipe galleries (PPGs), using cognitive backscatter technology and RF energy harvesting to address the challenges of battery dependency and communication conflicts in complex monitoring environments. A 3-stage system model is derived to use optimize energy acquisition, data transmission, and collision avoidance, which further uses techniques like range-level coupling and wake-up delay correlation analysis.
I have the following comments:
- 1) In the introduction, some missing and recent works are missing relating to RF power harvesting from intelligent surfaces perspective that can further help with wireless power transfer such as [R1]. Kindly revise.
- 2) Are there any trade-offs between communication collision probability \Delta p and energy consumption \Delta e ?
- 3) Explain computational complexity analysis that can be analyzed as function of the increased node density.
- 4) Clarify how minimizing collisions impacts energy efficiency and vice versa.
- 5) Specify the traditional communication mode used for comparison and its relevance to the proposed method.
- 6) Discuss any perturbed effect on the distance-energy level coupling and their impact on communication reliability.
- 7) What threshold is considered a success for data transmission? Also, how is "collision probability" measured or modeled in simulations?
- 8) There are som typos and weird phrasings like "...plays an important role to new power system which is safe and providing ..." which can be "... plays an important role in new power systems, which are safe and provide ..”. Please revise.
References
[R1] “Multi-Functional RIS for a Multi-Functional System: Integrating Sensing, Communication, and Wireless Power Transfer,” in IEEE Network, doi: 10.1109/MNET.2024.3482571
Comments on the Quality of English LanguageThe English could be improved to more clearly express the research.
Author Response
Comments 1: In the introduction, some missing and recent works are missing relating to RF power harvesting from intelligent surfaces perspective that can further help with wireless power transfer such as [R1]. Kindly revise.
References
[R1] “Multi-Functional RIS for a Multi-Functional System: Integrating Sensing, Communication, and Wireless Power Transfer,” in IEEE Network, doi: 10.1109/MNET.2024.3482571
Response 1: Thank you for pointing this out. We agree with this comment. Therefore, we cited this reference in the second paragraph of the introduction and summarized it. The location of the revision is on line 51 of the new manuscript.
Comments 2: Are there any trade-offs between communication collision probability \Delta p and energy consumption \Delta e ?
Response 2: Thank you for pointing this out. We agree with this comment. Analyzing from the working principle of the method in this paper, there is a certain correlation between the communication conflict probability \Delta p and the energy consumption \Delta e. This relationship can be summarized as follows: a lower collision probability corresponds to a lower system energy consumption. As the number of passive sensor nodes increases, the system collision probability rises, and the corresponding system energy consumption also increases. This relationship is also verified by sub - figures (a) and (c) of Figure 6. The core of this paper is to solve the problem of low - power communication for terminal sensors in the power status monitoring of PPGs. This does not affect the methodology and conclusions of this paper. Thanks to the expert's reminder, we will incorporate this content into our subsequent research.
Comments 3: Explain computational complexity analysis that can be analyzed as function of the increased node density.
Response 3: Thank you for pointing this out. We agree with this comment. Through analysis, it is found that in the method of this paper, for each additional node, the energy level of each node needs to be calculated separately at each simulated time moment. The computational complexity of this method is O(n). The location of the revision is on line 325 of the new manuscript.
Comments 4: Clarify how minimizing collisions impacts energy efficiency and vice versa.
Response 4: Thank you for pointing this out. We agree with this comment. Analyzing from the working principle of the method in this paper, there is a certain correlation between these two aspects. That is, reducing conflicts means that the probability of data transmission failure of passive sensor nodes will be reduced. Currently, the efficiency of radio frequency energy harvesting itself is limited. Passive sensor nodes use the stored energy for data transmission, but energy is wasted due to conflicts. Referring to the response to Comment 2, we will include this content in our subsequent research.
Comments 5: Specify the traditional communication mode used for comparison and its relevance to the proposed method.
Response 5: Thank you for pointing this out. We agree with this comment. The new manuscript explains the traditional communication mode ALOHA method used for comparison. The relevance between this method and the proposed method is that they are both communication methods that do not require an internal clock and are suitable for low - power communication scenarios. The location of the revision is in the first paragraph of Subsection 4.2.1.
Comments 6: Discuss any perturbed effect on the distance-energy level coupling and their impact on communication reliability.
Response 6: Thank you for pointing this out. We agree with this comment. There are many factors influencing the communication reliability of the distance energy level coupling analysis method, such as the distance between each passive sensor node and the radio frequency source, the environmental communication quality, the selection of the distance threshold and the energy level adjustment parameters, etc. In this paper, the distance threshold and energy level adjustment are selected as the key research objects. In the example part, the influence of the setting of the distance threshold and energy level adjustment parameters on the system collision probability is discussed. Considering that these factors do not affect the methodology and conclusions of this paper, how each parameter affects the system communication reliability will be included in our subsequent research.
Comments 7: What threshold is considered a success for data transmission? Also, how is "collision probability" measured or modeled in simulations?
Response 7: Thank you for pointing this out. We agree with this comment. "The collision probability" refers to the situation where more than one passive sensing node transmits data within a unit of time, which is considered a collision. In the simulation, "the collision probability" is a fraction. The denominator is the total number of observation time instants, and the numerator is the total number of time instants when collisions occur. The method for determining successful data transmission is supplemented in line 431 of Subsection 4.2.1.
Comments 8: There are som typos and weird phrasings like "...plays an important role to new power system which is safe and providing ..." which can be "... plays an important role in new power systems, which are safe and provide ..”. Please revise.
Response 8: Thank you for pointing this out. We agree with this comment. The new manuscript has been revised according to the expert's advice, and the revision is located at line 31 of the new manuscript.
Point 1: The English could be improved to more clearly express the research.
Response 1: Thank you for pointing this out. We agree with this comment. We have refined the English expressions in the manuscript.

Round 2
Reviewer 1 Report
Comments and Suggestions for Authors
The values used for transmission and reception gain (15dB) are too high. Such high gain values would correspond to very directional antennas, and therefore the angular dependence of their diagram should be taken into account. Only points within the antenna's beamwidth would receive a signal. Additionally, in the simulations, 20 dBm is taken as the transmitted power, but regulations limit the EIRP power, which consists of the product of the transmitter power and the gain of the transmitting antenna. Therefore, in the simulations, the regulations would be violated by emitting 20+15=35 dBm EIRP. I suggest reducing the gain to more reasonable values, such as a maximum of 6 dB, corresponding to a typical patch antenna used in these bands, but the transmission power should be limited to 20-6=14 dBm EIRP. Although it will likely not affect the final conclusion of the algorithm regarding the ALOHA access method, the simulation with more realistic parameters by modifying the powers will change the reading range and probably influence the comparison.
Author Response
Comments 1:The values used for transmission and reception gain (15dB) are too high. Such high gain values would correspond to very directional antennas, and therefore the angular dependence of their diagram should be taken into account. Only points within the antenna's beamwidth would receive a signal. Additionally, in the simulations, 20 dBm is taken as the transmitted power, but regulations limit the EIRP power, which consists of the product of the transmitter power and the gain of the transmitting antenna. Therefore, in the simulations, the regulations would be violated by emitting 20+15=35 dBm EIRP. I suggest reducing the gain to more reasonable values, such as a maximum of 6 dB, corresponding to a typical patch antenna used in these bands, but the transmission power should be limited to 20-6=14 dBm EIRP. Although it will likely not affect the final conclusion of the algorithm regarding the ALOHA access method, the simulation with more realistic parameters by modifying the powers will change the reading range and probably influence the comparison.
Response 1: Thank you for pointing this out. We agree with this comment.In the Case Study, we set the transmission power of the radio frequency (RF) source to 14 dBm, and the gains of the transmitting and receiving antennas to 6 dBm respectively. Considering that these values are smaller than the original ones, to ensure the effectiveness of RF energy harvesting, we reduced the distance between the RF source and the passive sensing nodes in the simulation. In the latest manuscript, we have controlled the distance between the RF source and each passive sensing node within 15 meters and re - simulated the example. Figures 5 - 8 have all been replaced, and the corresponding text in the article has also been revised. The revised parts in the text are marked in red.
Reviewer 2 Report
Comments and Suggestions for Authors
Authors have made good effort to address the reviewer's comments.
Author Response
Comments 1: Authors have made good effort to address the reviewer's comments.
Response 1: Thank you!
Reviewer 3 Report
Comments and Suggestions for Authors
This reviewer has no more comments.
Author Response
Comments 1: This reviewer has no more comments.
Response 1: Thank you!